

# Sodicity stress differently influences physiological traits and anti-oxidant enzymes in pear and peach cultivars

Anshuman Singh[1,2], Ashwani Kumar[1], Parbodh Chander Sharma[1], Raj Kumar[1] and Rajender Kumar Yadav[1]

[1] ICAR-Central Soil Salinity Research Institute, Karnal, Karnal, Haryana, India
[2] ICAR-Central Institute for Subtropical Horticulture, Lucknow, Uttar Pradesh, India

Corresponding author
Anshuman Singh,
anshumaniari@gmail.com

## ABSTRACT

**Background:** The growth and physiological responses to sodicity stress of pear and peach are poorly understood. Insights into how sodicity stress alters tree physiology remain vital to developing salt tolerant scion and rootstock cultivars.

**Methods:** The effects of sodicity stress (soil $pH_s$ ~8.8) on tree growth and physiological traits of field grown trees of pear cultivars Punjab Beauty and Patharnakh, and peach cultivars Partap and Shan-e-Punjab were recorded using standard procedures. Sodicity-induced changes in oxidative stressors, proline, anti-oxidant enzymes and leaf ions were measured to draw inferences.

**Results:** Sodicity-induced reductions in vegetative growth were particularly marked in Patharnakh pear and Partap peach compared with other cultivars. Although sodicity stress triggered a significant increase in leaf malondialdehyde (MDA) and hydrogen peroxide ($H_2O_2$), their levels relative to controls were much higher in peach than in pear; reflecting that peach suffered from greater oxidative stress. Interestingly, MDA and $H_2O_2$ levels did not seem to be deleterious enough to trigger proline-induced osmotic adjustment in pears. The activities of anti-oxidant enzymes strongly varied with the cultivar; specifically, the sodicity-induced increases in CAT and SOD activities were much higher in Punjab Beauty pear and Shan-e-Punjab peach. Principal Component Analysis revealed an explicit convergence between CAT and SOD activities in Punjab Beauty and Shan-e-Punjab cultivars in response to sodicity-induced oxidative stress. Correlation analysis revealed that leaf $Na^+$ strongly inhibited tree growth in peach than in pear. Leaf $K^+$ and proline were found to be the major osmolytes in sodicity-stressed pear and peach cultivars, respectively.

**Conclusions:** We have for the first time studied the effects of sodicity stress on important tree growth and physiological traits of commercially important pear and peach cultivars. Our findings revealed a marked suppressive effect of sodicity stress on tree growth in peach than in pear. The sodicity-induced upticks in leaf malondialdehyde, hydrogen peroxide and $Na^+$ seemed to induce proline-mediated osmotic adjustment in peach but not in pear. The overall better sodicity tolerance in pear compared to peach was ascribed to increased activities of anti-oxidant enzymes catalase and superoxide dismutase enzymes together with restricted $Na^+$ uptake and better leaf $K^+$ levels. Further investigations are needed to elucidate the effects of sodicity stress on genetic and transcriptional changes, and on fruit yield and quality.

# INTRODUCTION

Presently, about one billion hectare global land area is affected by salinity and sodicity problems to varying extents (*Hopmans et al., 2021*). Out of this, ~76 million hectare area is impacted by the human-induced secondary salinization and sodification (*Hossain, 2019*). The severity of the problem is evidenced by the fact that approximately 1.5 million hectares of land become unsuitable for agricultural production every year because of salinity and related problems (*Hossain, 2019*). Available evidence suggests that salt sensitive horticultural crops including pear (*Ahmad et al., 2022*) and peach (*Hernández et al., 2021*) are particularly adversely affected under saline and sodic conditions. In contrast to the commonly accepted threshold of 8.5 for the pH of saturated soil paste ($pH_s$), $pH_s$ >8.2 seems to be more realistic for classifying the soils as sodic under Indian conditions: soil $pH_s$ of 8.2 is often associated with an exchangeable sodium percentage (ESP) of 15; a limit above which the soils are generally considered to be sodic (*Abrol, Yadav & Massoud, 1988*). Generally, soil $pH_s$ of 8.5 roughly corresponds to ESP of ~50, high enough to suppress the crop growth (*Sharma et al., 2016*; *Abrol, Yadav & Massoud, 1988*). Although alkali salts such as sodium carbonate ($Na_2CO_3$) are often more detrimental to plant growth than the neutral salts such as sodium chloride (NaCl) (*Yang et al., 2009*), even in fairly salt tolerant crops (*Abbas et al., 2021*), alkali stress research continues to receive little attention. Sodic soils in the Trans-Gangetic Plains of India (study area) mostly have predominance of highly soluble $Na_2CO_3$ and $NaHCO_3$ salts, and are thus prone to the abrupt increases in the soil pH (*Mandal, 2012*).

In sodic soils, excessive $Na^+$ causes the clay dispersion, surface crusting and deterioration in the soil physical properties (*Qadir et al., 2007*). Besides poor physical properties, high pH, osmotic and ionic stresses, and nutrient deficiencies are other limitations to plant growth in the sodic soils (*Qadir & Schubert, 2002*). Additionally, calcium carbonate ($CaCO_3$) concretions in the sub-soil also hamper plant establishment (*Sharma et al., 2016*). Considerable spatial variations in soil pH are also frequently seen in sodic soils; the more sodic parts of the field are often less congenial for crop growth (*Samra, Singh & Sharma, 1988*). In fruit crops, the adverse effects of sodicity stress on plant growth (*Saxena & Gupta, 2006*; *Krishnamoorthy, 2009*) are ascribed to altered plant water relations (*Li et al., 2020*), reduced levels of photosynthetic pigments (*Krishnamoorthy, 2009*; *Li et al., 2020*), lipid peroxidation and oxidative stress (*Ahmad et al., 2014*), and ionic stress (*Singh et al., 2018b*). Sodicity-stressed plants accumulate osmolytes such as proline for osmotic adjustment (*Krishnamoorthy, 2009*; *Ahmad et al., 2014*; *Singh et al., 2018a*), and also activate the antioxidant enzymes for scavenging the free radicals (*Ahmad et al., 2014*).

Although sodic soils display noticeable improvements in the physico-chemical properties following amendment application and salt leaching with the fresh water, such improvements are mostly transient and confined to the top soil (<15 cm) (*Sharma & Singh, 2019*) such that sub-soil constraints continue to persist (*Kumar et al., 2019*). Under

such conditions, agronomic practices such as planting into amendment-treated auger-holes often give better results (*Gill & Abrol, 1991*; *Saxena & Gupta, 2006*). As a majority of fruit crops are mostly highly sensitive to salinity and sodicity stresses (*Singh et al., 2018a*), development of salt tolerant scion and rootstock cultivars is absolutely essential to sustain the fruit production in salt-affected soils (*Mahmoud et al., 2020*). Although salt tolerance is a complex polygenic trait, and is greatly influenced by the genetic and environmental cues (*Flowers, 2004*), there exists ample genotypic variation that needs to be explored for identifying the salt tolerant genotypes (*Mahmoud et al., 2020*; *Singh et al., 2018a*).

Although some studies have shown the adverse effects of salinity stress on tree growth and physiology of pears (*Myers et al., 1995*; *Musacchi, Quartieri & Tagliavini, 2006*; *Oron et al., 2002*) and peaches (*Boland, Mitchell & Jerie, 1993*; *Karakas, Bianco & Rieger, 2000*; *Soliman, Abo-Ogiela & El-Saedony, 2017*), their responses to sodicity stress remain elusive; excluding some preliminary observations that high soil pH and the related sub-soil constraints may suppress the plant growth (*Elkins, Bell & Einhorn, 2012*; *Abd-Elmegeed, Nabeel & Nasr, 2013*; *Mestre et al., 2015*; *Tagliavini, Masia & Quartieri, 1995*). However, the soil pH levels in these studies were rather low to draw reasonable inferences, and the plausible physiological changes accounting for the reduced plant growth were also not investigated. Plant responses to salt stress often strikingly vary with the experimental conditions; for instance, depending on experimental conditions, pears are either highly sensitive (*Ebert, 1999*) or moderately tolerant (*Musacchi, Quartieri & Tagliavini, 2006*) to the salinity stress. Interestingly, most of the aforementioned studies had used NaCl as the sole salinizing agent in relatively controlled short-term experiments; the results may be altogether different when a different salt is used to induce the salt stress (*Grieve, Grattan & Maas, 2012*). While an interplay among the oxidative stressors such as malondialdehyde, osmolytes, anti-oxidant enzymes and the leaf ions is known to greatly influence the plant responses to salinity stress in pears and peaches (*Erturk et al., 2007*; *Dejampour et al., 2012*; *Wu & Zou, 2009*; *Yousefi, Naseri & Zaare-Nahandi, 2019*), such responses remain uninvestigated under sodic conditions. Insights from other fruit crops (*e.g.*, *Malus halliana* (*Jia et al., 2019*) and *Morus alba* (*Hui-Hui et al., 2019*)) also suggest that plant physiological responses to salinity and sodicity stresses remarkably vary with each other: the effects of salinity cannot be used as a reliable proxy for the effects of sodicity.

The subtropical areas in the Trans-Gangetic Plains of India (study region) usually have a frost free spring and adequate chilling availability for the cultivation of low chill cultivars of temperate fruits such as pear and peach. Pear cultivars Patharnakh (*Pyrus pyrifolia* (Burm F.) Nakai) and Punjab Beauty (*Pyrus communis* L.) (*Singh et al., 2015*), and peach cultivars Partap and Shan-e-Punjab (*Prunus persica* (L.) Batsch) (*Bhatnagar & Kaul, 2002*) are particularly suitable for cultivation in this area given their commercial potential, low chilling requirement and the ability to tolerate the high summer temperatures typical of the region. To our knowledge, systematic studies have not yet been carried out to evaluate the effects of sodicity stress on tree growth and physiological relations of pears and peaches. This study intended to delineate sodicity-induced changes in tree growth, leaf oxidative stress markers, proline, enzymatic anti-oxidants and leaf ions in pear and peach

cultivars, and how these changes influence the overall cultivar-specific tree responses to sodicity stress.

## MATERIALS AND METHODS

### Location

The experiment was conducted between January, 2015 and April, 2019 at the experimental farm of Indian Council of Agricultural Research-Central Soil Salinity Research Institute, Karnal, India (29°42′20.6″N, 76°57′19.80″E, 243 m above mean sea level). The region has a semi-arid subtropical climate with hot summers and dry winters. The long-term average annual rainfall is ~700 mm. A sodic field was used to evaluate the pear and peach cultivars.

### Experimental material

The pear cultivars Punjab Beauty (*Pyrus communis* L.) and Patharnakh (*Pyrus pyrifolia* (Burm F.) Nakai) grafted on Kainth (*Pyrus pashia* Buch.-Ham. Ex D. Don); and peach cultivars Partap and Shan-e-Punjab (*Prunus persica* (L.) Batch) on Sharbati rootstock (*P. persica* (L.) Batch) were evaluated. The bare root plants were planted on 17[th] January 2015 in auger-holes (diameter 20 cm, depth 120 cm) in a sodic field, keeping the graft joint ~15 cm above the surface. The auger-holes were refilled with a mixture of the original soil and 5 kg of gypsum ($Ca_2SO_4.2H_2O$) before planting for better plant establishment (*Sharma, Chaudhari & Singh, 2014*). The square system of planting was used, with between- and within-row spacings of 6 m each in both pears (*Sangwan et al., 2015*) and peaches (*Kanwar & Singh, 2004*). Trees were trained to the modified leader system, and the recommended management practices were followed for better tree growth. Irrigation water was applied through 1 m wide channels. Plants were irrigated at the weekly intervals during summer months and at fortnightly intervals during July-August.

### Treatments

Soil samples were collected from 24 random points of the field, from four depths (0–15, 15–30, 30–60 and 60–100 cm) using an auger. After air drying, the samples were ground and sieved (2.0 mm sieve) for determining the pH of saturated soil paste ($pH_s$) and soil saturation extract electrical conductivity ($EC_e$) using digital pH and conductivity meters (Eutech, Singapore, Asia). Based on soil analysis, the experimental plants were grouped into control (mean soil $pH_s$ = 8.22, $EC_e$ = 0.71 dS m$^{-1}$) and sodic ($pH_s$ = 8.80, $EC_e$ = 0.94 dS m$^{-1}$) treatments for recording the observations. Both soil $pH_s$ and $EC_e$ increased with depth, and there were significant differences between control and sodicity treatments ($pH_s$ $F = 41.94$, $p < 0.001$; $EC_e$ $F = 27.52$, $p < 0.001$) (Table S1). The groundwater used in irrigation had the following composition: electrical conductivity- 0.65 dS m$^{-1}$, pH-8.04, Na$^+$-2.41 me L$^{-1}$, K$^+$-0.15 me L$^{-1}$, Ca$^{2+}$ + Mg$^{2+}$-4.17 me L$^{-1}$, Cl$^-$-0.98 me L$^{-1}$ and HCO$^-_3$-4.31 me L$^{-1}$.

### Tree growth

Four trees of each pear and peach cultivar representing the control and sodic treatments were randomly tagged for recording the observations. Tree height and trunk diameter were

measured during the last week of April, 2019. Trunk diameter readings, recorded using a digital Vernier caliper (Mitutoyo, Kawasaki, Kanagawa, Japan) 15 cm above the graft union, were converted into trunk cross sectional area (TCSA) by the formula: TCSA = $\pi(d/2)^2$; where d = mean of east-west and north-south trunk diameters. The canopy volume (CV) was computed by the formula: CV = $(w^2 \times h)/2$; where w = canopy diameter in east-west and north-south directions; and h = tree height.

## Leaf physiological traits

The fully expanded leaves from the middle of the shoots were collected from all the directions of each replicate tree ($n = 4$), pooled, packed in zip lock bags inside an ice-box, and immediately brought to the laboratory. The total leaf chlorophyll (TC) was estimated by overnight incubation of 200 mg chopped leaves in 80% acetone (*Arnon, 1949*) using a spectrophotometer (UV 3200, Lab India Analytical, India). Lipid peroxidation, in terms of malondialdehyde (MDA) content, was estimated by the method of *Heath & Packer (1968)*. A total of 1 g of leaf tissue was homogenized in 5 ml of 0.1 percent trichloroacetic acid (TCA, w/v) and centrifuged at 8,000 g for 15 min. Supernatant (1 ml) was precipitated in 4 ml of 20% (w/v) TCA containing 0.5% (w/v) 2-thiobarbituric acid. The reaction mixture was heated in a water bath (at 95 °C for 30 min) with constant stirring and quickly cooled in the ice bath. Sample was centrifuged at 8,000×$g$ for 10 min, absorbance of the supernatant was measured at 532 nm against distilled water, and the value for non-specific absorption at 600 nm was subtracted. The MDA concentration was calculated by using the molar extinction coefficient of 155 mM$^{-1}$ cm$^{-1}$. Hydrogen peroxide ($H_2O_2$) content was estimated using the procedure described in *Sinha (1972)*. The leaf tissue was macerated in 5 ml of ice cold 0.01 M phosphate buffer (pH 7.0) and centrifuged at 8,000 g for 10 min. 1.95 ml of 0.01 M potassium phosphate buffer (pH 7.0) followed by 5% potassium dichromate (2 ml) and glacial acetic acid (1:3; v/v) were added to 50 µl supernatant. The sample tubes were kept in the boiling water bath for 10 min and then cooled. The absorbance was read at 570 nm against the reagent blank without sample extract and the $H_2O_2$ content was calculated from the standard calibration curve (10 to 160 µmol of $H_2O_2$). Proline was extracted using 200 mg leaf tissue homogenized in 10 ml of 3% sulphosalicyclic acid (*Bates, Waldren & Teare, 1973*). Two ml of extract was reacted with 2 ml each of acid-ninhydrin and glacial acetic acid for 1 h at 100 °C, and the reaction was terminated in water bath. The reaction mixture was extracted with 4 ml toluene, mixed vigorously and the chromophore containing toluene was aspirated from the aqueous phase. The absorbance was measured at 520 nm using toluene as blank.

## Anti-oxidant enzymes

Fresh leaf samples (~250 mg) were homogenized in 0.1 M phosphate buffer (pH 7.5) containing 5% polyvinyl polypyrrolidone (w/v), 1 mM ethylenediamine tetraacetic acid, and 10 mM b-mercapto-ethanol for assaying the ascorbate peroxidase (APX, EC 1.11.1.11) and superoxide dismutase (SOD, EC 1.15.1.1) activities. The APX activity was assayed as described in *Nakano & Asada (1981)*; enzyme activity was calculated using extinction coefficient of 2.8 mM$^{-1}$ cm$^{-1}$. The SOD assay was performed following the methodology

given in *Beauchamp & Fridovich (1971)*. The absorbance of the solution was measured at 560 nm with a UV-VIS spectrophotometer (UV 3200, Lab India Analytical, Maharashtra, India). One unit of SOD was calculated as the amount of enzyme required to inhibit the photo-reduction of one mmol of nitroblue tetrazolium. Catalase (CAT, EC 1.11.1.6) and peroxidase (POX, EC 1.11.1.7) were extracted in 0.01 M phosphate buffer (pH 7.5) with 3% polyvinyl polypyrrolidone (w/v) by homogenizing the fresh leaf tissue (1.0 g). The homogenate was centrifuged at 4 °C for 15 min at 10,000×$g$ and the clear supernatant was used for the assay. The CAT activity was determined as the disappearance of $H_2O_2$ at 240 nm (25 °C) for 1 min (*Aebi, 1984*). The POX was assayed by determining the rate of guaiacol oxidation in the presence of $H_2O_2$ at 470 nm (*Rao et al., 1998*). One unit of POX activity was defined as the amount of enzyme required to oxidize one nmol of guaiacol $min^{-1}$ $ml^{-1}$.

## Leaf Na$^+$ and K$^+$

The leaf samples were first dried to a constant weight at 60 °C in a hot air oven (Narang Scientific Works, Delhi, India). The samples were then finely ground using a hammer mill. One hundred mg of the powdered sample was digested in the di-acid (nitric acid ($HNO_3$) and perchloric acid ($HClO_4$); 3:1) mixture for determining the Na$^+$ and K$^+$ contents (mg $g^{-1}$ DW) using a flame photometer (Systronics India, Gujarat, India).

## Statistical analysis

The experiment was laid out in a randomized block design. The main and interaction effects of independent factors (treatment and cultivar) on each dependent variable were examined by a two-way Analysis of Variance (ANOVA). The assumptions of equality of variances (Levene's test) and normality (quantile-quantile plot) were checked prior to ANOVA, and some variables (tree height, proline and peroxidase in case of peach data) were log-transformed to improve the ANOVA assumptions. The effect size measure omega squared ($\omega^2$) was computed to estimate the variance in the response variable(s) accounted for by the explanatory variables. The use of sample size independent effect size measures such as $\omega^2$ improves the statistical power by reducing the risk of Type II error (*Altman, 2000*). Tukey's test ($p < 0.05$) was used for mean comparisons (JASP v. 0.15). The data are expressed as mean ($n = 4$) ± standard deviation (SD). Principal component analysis (PCA) (Bartlett's test of sphericity, $p < 0.001$) was applied to reduce the dimensionality and to detect the key patterns in data (Jamovi v. 2.2). Pearson's bivariate correlations between the measured traits were computed (*Julkowska et al., 2019*).

## RESULTS

### Analysis of variance

The results for the analysis of variance (ANOVA) revealed the strong repressive effects of sodicity stress on tree growth and physiological traits in both pear and peach. Although sodicity-induced reductions in tree height (TH), trunk cross sectional area (TCSA) and canopy volume (CV) were highly significant ($p < 0.001$) in both the crops, a perusal of the effect-size measure ($\omega^2$) values implied that TCSA was far less sensitive to sodicity stress

than were both TH and CV; regardless of the crop (Table 1). Likewise, based on $\omega^2$ values, a more adverse effect of sodicity stress was apparent on peach than on pear growth. The $\omega^2$ values were low-to-moderate (<0.600) for the most leaf physiological traits, but relatively high (>0.700) for leaf proline, ascorbate peroxidase (APX) and $Na^+$ in pear. This suggested that explanatory variable (*i.e.*, sodicity stress) accounted for a reasonably high variance in leaf proline, APX and $Na^+$. Pear cultivars differed markedly with each other for all the traits except leaf proline, APX and $Na^+/K^+$ ratio (Table 1). The sodicity-triggered increases in leaf malondialdehyde (MDA) and hydrogen peroxide ($H_2O_2$) levels were far greater in peach compared to pear in terms of $\omega^2$ values. This, together with more or less similar values of $\omega^2$ for the leaf proline, implied the greater sensitivity to oxidative stress of peaches than of pears. Contrarily, the values of $\omega^2$ evinced moderate-to-strong increases in APX, peroxidase (POX) and catalase (CAT) activities in pear but not in peach; while upregulation in the superoxide dismutase (SOD) activity was quite similar in both the cases. This again indicated a better anti-oxidant system to cope with the free radicals in pear than in peach (Table 1).

## Tree growth

There were strong differences between the cultivars for sodicity-induced reductions in tree growth. For instance, the reductions in trunk cross sectional area (TCSA) and canopy volume (CV) relative to controls were much lower in Punjab Beauty pear (16.49% and 44.50%, respectively) than in Patharnakh (41.89% and 69.28%, respectively) (Table 2). Sodicity stressed Partap peach trees displayed much higher reductions in tree height (TH, 36.24%), TCSA (74.28%) and CV (90.33%) than corresponding decreases of 29.23%, 11.94% and 58.58% in the cultivar Shan-e-Punjab (Table 2).

## Leaf physiological traits

Sodicity stress caused appreciable reductions in total leaf chlorophyll (TC) in pear (Punjab Beauty-26.95%, Patharnakh-21.43%). Leaf malondialdehyde (MDA), hydrogen peroxide ($H_2O_2$) and proline levels were invariably higher in sodic ($pH_s$ 8.8) than in control ($pH_s$ 8.2) treatment. Leaf MDA increased marginally (9.61%) in Punjab Beauty and moderately (19.92%) in Patharnakh (Table 3). Both the cultivars showed identical increases (~12.0%) in leaf $H_2O_2$. Leaf proline accumulation in sodic treatment ($pH_s$ 8.8) was considerably higher in Punjab Beauty (33.86%) compared to Patharnakh (20.15%). Partap and Shan-e-Punjab peaches had 30.80 and 18.09% less TC, respectively, in sodic soils ($pH_s$ 8.8) than in controls ($pH_s$ 8.2) (Table 3).

## Leaf anti-oxidant enzymes

Sodicity-triggered increases in ascorbate peroxidase (APX) and catalase (CAT) activities relative to controls were more pronounced in pear cultivar Patharnakh (34.49% and 35.97%, respectively) than in Punjab Beauty (28.44% and 25.16%, respectively) (Table 4). Contrarily, peroxidase (POX) and superoxide dismutase (SOD) activities were 2.6- and 1.4-folds higher, respectively, in sodicity-stressed Punjab Beauty leaves than in Patharnakh (Table 4). However, in absolute terms, only POX activity was higher in sodicity-stressed

**Table 1** Analysis of Variance (ANOVA) for different traits in pear and peach.

| Trait | Source | F | p | $\omega^2$ | F | p | $\omega^2$ |
|---|---|---|---|---|---|---|---|
| | | Pear | | | Peach | | |
| Tree height (m) | Treatment (T) | 83.55 | <0.001 | 0.354 | 218.76 | <0.001 | 0.902 |
| | Cultivar (C) | 134.00 | <0.001 | 0.571 | 5.95 | 0.031 | 0.021 |
| | T × C | 2.35 | 0.151* | 0.006 | 3.62 | 0.081* | 0.011 |
| Trunk cross sectional area (cm²) | Treatment (T) | 50.98 | <0.001 | 0.203 | 468.82 | <0.001 | 0.379 |
| | Cultivar (C) | 179.72 | <0.001 | 0.724 | 533.37 | <0.001 | 0.431 |
| | T × C | 2.98 | 0.110* | 0.008 | 219.17 | <0.001 | 0.177 |
| Canopy volume (m³) | Treatment (T) | 38.93 | <0.001 | 0.606 | 411.53 | <0.001 | 0.846 |
| | Cultivar (C) | 10.20 | 0.008 | 0.147 | 53.81 | <0.001 | 0.109 |
| | T × C | 0.50 | 0.495* | 0.000 | 7.15 | 0.020 | 0.013 |
| Total leaf chlrophyll (mg/g FW) | Treatment (T) | 36.52 | <0.001 | 0.509 | 46.78 | <0.001 | 0.366 |
| | Cultivar (C) | 18.39 | 0.001 | 0.249 | 55.54 | <0.001 | 0.436 |
| | T × C | 1.86 | 0.197* | 0.012 | 9.69 | 0.009 | 0.070 |
| Malondialdehyde (nmoles/g FW) | Treatment (T) | 30.65 | <0.001 | 0.376 | 81.82 | <0.001 | 0.818 |
| | Cultivar (C) | 32.93 | <0.001 | 0.405 | 3.96 | 0.070* | 0.030 |
| | T × C | 2.20 | 0.164* | 0.015 | 0.03 | 0.861* | 0.000 |
| Hydrogen peroxide (mmoles/g FW) | Treatment (T) | 84.27 | <0.001 | 0.219 | 101.26 | <0.001 | 0.772 |
| | Cultivar (C) | 281.57 | <0.001 | 0.739 | 15.27 | 0.002 | 0.110 |
| | T × C | 0.95 | 0.349* | 0.000 | 0.32 | 0.580* | 0.000 |
| Proline (mg/g FW) | Treatment (T) | 40.49 | <0.001 | 0.703 | 85.00 | <0.001 | 0.853 |
| | Cultivar (C) | 0.42 | 0.530* | 0.000 | 0.44 | 0.518* | 0.000 |
| | T × C | 2.28 | 0.157* | 0.023 | 0.06 | 0.814* | 0.000 |
| Ascorbate peroxidase (units/g FW) | Treatment (T) | 147.59 | <0.001 | 0.906 | 28.11 | <0.001 | 0.092 |
| | Cultivar (C) | 0.42 | 0.527* | 0.000 | 252.51 | <0.001 | 0.856 |
| | T × C | 0.85 | 0.374* | 0.000 | 0.21 | 0.652* | 0.000 |
| Peroxidase (units/g FW) | Treatment (T) | 167.45 | <0.001 | 0.500 | 130.39 | <0.001 | 0.072 |
| | Cultivar (C) | 129.46 | <0.001 | 0.386 | 1618.96 | <0.001 | 0.898 |
| | T × C | 23.24 | <0.001 | 0.067 | 39.18 | <0.001 | 0.021 |
| Catalase (units/g FW) | Treatment (T) | 161.13 | <0.001 | 0.421 | 197.25 | <0.001 | 0.116 |
| | Cultivar (C) | 205.79 | <0.001 | 0.539 | 1456.33 | <0.001 | 0.860 |
| | T × C | 1.040e−4 | 0.992* | 0.000 | 24.82 | <0.001 | 0.014 |
| Superoxde dismutase (units/g FW) | Treatment (T) | 88.32 | <0.001 | 0.542 | 107.21 | <0.001 | 0.554 |
| | Cultivar (C) | 55.54 | <0.001 | 0.338 | 71.48 | <0.001 | 0.368 |
| | T × C | 4.35 | 0.059* | 0.021 | 0.01 | 0.911* | 0.000 |
| Leaf Na⁺ (mg/g DW) | Treatment (T) | 159.87 | <0.001 | 0.745 | 423.82 | <0.001 | 0.730 |
| | Cultivar (C) | 38.69 | <0.001 | 0.177 | 121.42 | <0.001 | 0.208 |
| | T × C | 1.80 | 0.205* | 0.004 | 21.05 | <0.001 | 0.035 |
| Leaf K⁺ (mg/g DW) | Treatment (T) | 109.07 | <0.001 | 0.590 | 252.43 | <0.001 | 0.144 |
| | Cultivar (C) | 58.23 | <0.001 | 0.312 | 1407.86 | <0.001 | 0.806 |
| | T × C | 2.96 | 0.111* | 0.011 | 71.96 | <0.001 | 0.041 |
| Leaf Na⁺/K⁺ ratio | Treatment (T) | 201.17 | <0.001 | 0.871 | 651.86 | <0.001 | 0.620 |

| Table 1 (continued) | | | | | | | |
|---|---|---|---|---|---|---|---|
| **Trait** | **Source** | **F** | ***p*** | $\omega^2$ | **F** | ***p*** | $\omega^2$ |
| | | **Pear** | | | **Peach** | | |
| | Cultivar (C) | 0.38 | 0.549* | 0.000 | 360.43 | <0.001 | 0.342 |
| | T × C | 15.25 | 0.002 | 0.062 | 25.19 | <0.001 | 0.023 |

Note:
* Non-significant effect ($p > 0.05$).

**Table 2 Mean comparisons for tree growth parameters in pear and peach cultivars.**

| Cultivar | Treatment | Tree height (m) | Trunk cross sectional area (cm²) | Canopy volume (m³) |
|---|---|---|---|---|
| **Pear** | | | | |
| Punjab beauty | Control | 3.86 ± 0.20a | 36.76 ± 2.63a | 1.91 ± 0.21a |
| | Sodic | 3.23 ± 0.12b | 30.70 ± 2.64b | 1.06 ± 0.19bc |
| Patharnakh | Control | 3.03 ± 0.20b | 23.68 ± 1.87c | 1.53 ± 0.53ab |
| | Sodic | 2.14 ± 0.13c | 13.76 ± 1.62d | 0.47 ± 0.08d |
| **Peach** | | | | |
| Partap | Control | 4.25 ± 0.31a | 231.17 ± 9.61b | 28.96 ± 2.90a |
| | Sodic | 2.71 ± 0.11b | 59.45 ± 2.62c | 2.80 ± 0.18b |
| Shan-e-Punjab | Control | 4.31 ± 0.14a | 270.22 ± 12.0a | 34.28 ± 3.09a |
| | Sodic | 3.05 ± 0.18b | 237.96 ± 10.57b | 14.20 ± 1.68b |

Note:
Control and sodic treatments denote significantly different ($p < 0.001$) pH$_s$ levels of ~8.2 and 8.8, respectively. Each data value represents mean ($n = 4$) ± SD.

**Table 3 Mean comparisons for leaf physiological traits in pear and peach cultivars.**

| Cultivar | Treatment | TC (mg g⁻¹ FW) | MDA (nmoles g⁻¹ FW) | H₂O₂ (mmoles g⁻¹ FW) | Proline (mg g⁻¹ FW) |
|---|---|---|---|---|---|
| **Pear** | | | | | |
| Punjab beauty | Control | 1.41 ± 0.10a | 8.95 ± 0.25b | 130.41 ± 1.99b | 3.78 ± 0.43b |
| | Sodic | 1.03 ± 0.10bc | 9.81 ± 0.35a | 146.27 ± 3.23a | 5.06 ± 0.25a |
| Patharnakh | Control | 1.12 ± 0.05a | 7.43 ± 0.22c | 105.74 ± 4.32d | 3.92 ± 0.28b |
| | Sodic | 0.88 ± 0.14c | 8.91 ± 0.69b | 118.55 ± 2.43c | 4.71 ± 0.31a |
| **Peach** | | | | | |
| Partap | Control | 1.56 ± 0.11a | 7.74 ± 0.36b | 138.12 ± 4.28c | 4.06 ± 0.11b |
| | Sodic | 1.08 ± 0.15b | 10.58 ± 0.94a | 175.57 ± 8.77a | 5.31 ± 0.18a |
| Shan-e-Punjab | Control | 1.05 ± 0.05bc | 7.04 ± 0.11b | 153.89 ± 9.19b | 4.16 ± 0.19b |
| | Sodic | 0.86 ± 0.05c | 10.0 ± 0.78a | 187.33 ± 4.32a | 5.39 ± 0.50a |

Note:
Control and sodic treatments denote significantly different ($p < 0.001$) pH$_s$ levels of ~8.2 and 8.8, respectively. Each data value represents mean ($n = 4$) ± SD. TC- total leaf chlorophyll, MDA-malondialdehyde, $H_2O_2$-hydrogen peroxide.

Patharnakh while both CAT and SOD activities much higher in Punjab Beauty. In case of peach, the sodicity-triggered upregulation in leaf APX and POX activities relative to controls were markedly higher in Shan-e-Punjab (29.41% and 22.99%, respectively) than

**Table 4 Mean comparisons for leaf anti-oxidant enzymes (units g$^{-1}$ FW) in pear and peach cultivars.**

| Cultivar | Treatment | Ascorbate peroxidase | Peroxidase | Catalase | Superoxide dismutase |
|---|---|---|---|---|---|
| Pear | | | | | |
| Punjab Beauty | Control | 14.10 ± 0.53b | 183.68 ± 5.80c | 6.20 ± 0.31b | 64.09 ± 1.88b |
| | Sodic | 18.11 ± 0.75a | 224.07 ± 4.92b | 7.76 ± 0.24a | 80.62 ± 4.40a |
| Patharnakh | Control | 13.54 ± 0.68b | 220.47 ± 3.47b | 4.42 ± 0.31c | 56.23 ± 1.29c |
| | Sodic | 18.21 ± 0.86a | 238.84 ± 3.74a | 6.01 ± 0.25b | 66.84 ± 2.08b |
| Peach | | | | | |
| Partap | Control | 22.15 ± 1.38b | 307.41 ± 5.85b | 3.93 ± 0.17d | 46.88 ± 1.31c |
| | Sodic | 25.14 ± 1.46a | 326.60 ± 5.42a | 5.22 ± 0.43c | 54.47 ± 1.55b |
| Shan-e-Punjab | Control | 12.07 ± 1.01d | 178.22 ± 6.93d | 6.93 ± 0.37b | 53.00 ± 1.57b |
| | Sodic | 15.62 ± 1.01c | 219.20 ± 5.92c | 7.51 ± 0.41a | 60.84 ± 1.54a |

Note:
Control and sodic treatments denote significantly different ($p < 0.001$) pH$_s$ levels of ~8.2 and 8.8, respectively. Each data value represents mean ($n = 4$) ± SD.

in Partap (13.49% and 6.24%, respectively). Conversely, CAT and SOD activities were more noticeable in the cultivar Partap (38.82% and 16.19%, respectively) compared to the cultivar Shan-e-Punjab (8.37% and 14.79%, respectively). Nonetheless, Shan-e-Punjab significantly outperformed Partap for the absolute leaf CAT and SOD levels in the sodic soils (pH$_s$ 8.8) (Table 4).

## Leaf Na$^+$ and K$^+$

As expected, sodicity stress caused an increase in leaf Na$^+$ and a decrease in K$^+$, regardless of the cultivar. Pear cultivars Punjab Beauty and Patharnakh displayed considerably higher leaf Na$^+$ (43.91% and 74.57%, respectively) in sodic (pH$_s$ 8.8) than in control (pH$_s$ 8.2) soils. In contrast, leaf K$^+$ declined significantly in sodicity stressed Punjab Beauty (17.76%) and Patharnakh (28.69%) trees (Table 5). In sodic soils (pH$_s$ 8.8), peach cultivars Partap and Shan-e-Punjab had 71.68% and 58.24% more leaf Na$^+$, respectively, and 31.97% and 24.21% less K$^+$, respectively (Table 5).

## Principal component analysis

The principal component analysis (PCA) was quite efficient in reducing the dimensionality: the first two principal components (PCs) (Eigen value >1.0) alone explained 90.95% of the cumulative variance in data in pear (Table S2; Fig. 1), and 94.61% in peach (Table S2; Fig. 2). In case of pear, leaf Na$^+$, K$^+$ and Na$^+$/K$^+$ ratio alongside proline, APX and CV were the highly weighted variables on PC1; and TH, TCSA, HP, MDA and CAT on PC2 (Table S2). Likewise, for peach, TH, HP, proline and Na$^+$/K$^+$ ratio were the best represented variables on PC1, and TCSA, APX, POX, CAT and K$^+$ on PC2 (Table S2). Interestingly, PCA achieved a clear-cut discrimination for the cultivar- and treatment-specific effects in data: while PC1 unambiguously distinguished the control and sodicity treatments, PC2 clearly separated the tested cultivars from each other (Figs. 1 and 2). The PCA also unveiled some interesting patterns in data. For pear, tight clustering of tree growth attributes, K and TC in the upper left quadrant implied a straightforward role of leaf K$^+$ in osmotic adjustment in the sodicity-stressed pears (Fig. 1). Likewise, there was

**Table 5  Mean comparisons for leaf Na⁺ and K⁺ and Na⁺: K⁺ ratio in pear and peach cultivars.**

| Cultivar | Treatment | $Na^+$ (mg/g DW) | $K^+$ (mg/g DW) | $Na^+$: $K^+$ ratio |
|---|---|---|---|---|
| Pear | | | | |
| Punjab Beauty | Control | 0.82 ± 0.07c | 9.74 ± 0.52a | 0.08 ± 0.01b |
| | Sodic | 1.18 ± 0.09a | 8.01 ± 0.31b | 0.15 ± 0.01a |
| Patharnakh | Control | 0.59 ± 0.05d | 8.54 ± 0.63b | 0.07 ± 0.01b |
| | Sodic | 1.03 ± 0.07b | 6.09 ± 0.45c | 0.17 ± 0.02a |
| Peach | | | | |
| Partap | Control | 1.13 ± 0.10c | 14.42 ± 0.59a | 0.08 ± 0.01d |
| | Sodic | 1.94 ± 0.10a | 9.81 ± 0.25b | 0.20 ± 0.01b |
| Shan-e-Punjab | Control | 0.91 ± 0.04d | 5.70 ± 0.37c | 0.16 ± 0.01c |
| | Sodic | 1.44 ± 0.09b | 4.32 ± 0.26d | 0.33 ± 0.02a |

**Note:**
Control and sodic treatments denote significantly different ($p < 0.001$) $pH_s$ levels of ~8.2 and 8.8, respectively. Each data value represents mean ($n = 4$) ± SD.

an apparent cultivar-specific upregulation in CAT and SOD activities in response to the oxidative ($H_2O_2$ and MDA) and ionic ($Na^+$) stresses in Punjab Beauty trees; increased POX activity alone seemed to have alleviated salt stress in cultivar Patharnakh (Fig. 1). In the case of peach, leaf $K^+$ did not have any obvious effect in alleviating the salt stress. Contrarily, osmoregulation through proline apparently increased in response to increasing HP, MDA and $Na^+$ levels. Of the enzymatic anti-oxidants, APX and POX were rather specific to the cultivar Partap, and CAT and SOD to the cultivar Shan-e-Punjab (Fig. 2).

### Correlation analysis

In pear, tree growth traits and total leaf chlorophyll (TC) had strong positive correlations with leaf $K^+$ ($r > 0.800$, $p = 0.000$). Expectedly, leaf $Na^+/K^+$ ratio had moderate-to-strong negative correlations with TH ($r = -0.623$, $p = 0.010$), TC ($r = -0.602$, $p = 0.014$) and CV ($r = -0.788$ $p = 0.000$) (Table S3, Fig. 3). A strong positive correlation between leaf MDA and $H_2O_2$ ($r = 0.914$, $p = 0.000$) reflected their synergistic adverse effects on pear tree growth and physiology. Leaf proline had moderate positive correlations with MDA ($r = 0.578$, $p = 0.019$) and $H_2O_2$ ($r = 0.519$, $p = 0.039$). Similarly, strong positive relationships ($r > 0.800$, $p = 0.000$) were found between antioxidants (CAT and SOD) and oxidative stress indicators (MDA and $H_2O_2$) (Table S3, Fig. 3). In peach, leaf $K^+$ had strong positive correlations with TC ($r = 0.887$, $p = 0.000$), APX ($r = 0.732$, $p = 0.001$) and POX ($r = 0.791$, $p = 0.000$), strong negative correlations with CAT ($r = -0.987$, $p = 0.000$) and SOD ($r = -0.839$, $p = 0.000$), and a moderate negative correlation with $H_2O_2$ ($r = -0.655$, $p = 0.006$) (Table S4, Fig. 4). In contrast to pear, elevated leaf $Na^+$ had strong inhibitory effects ($r > -0.880$, $p = 0.000$) on tree growth in peach. The adverse effects of both MDA and HP on peach tree growth were also apparent ($r > -0.800$) (Table S4, Fig. 4).

### DISCUSSION

Sodicity stress is a serious constraint on pear and peach productivity (*Elkins, Bell & Einhorn, 2012*; *Mestre et al., 2015*). The observation that sodicity stress impairs pear and

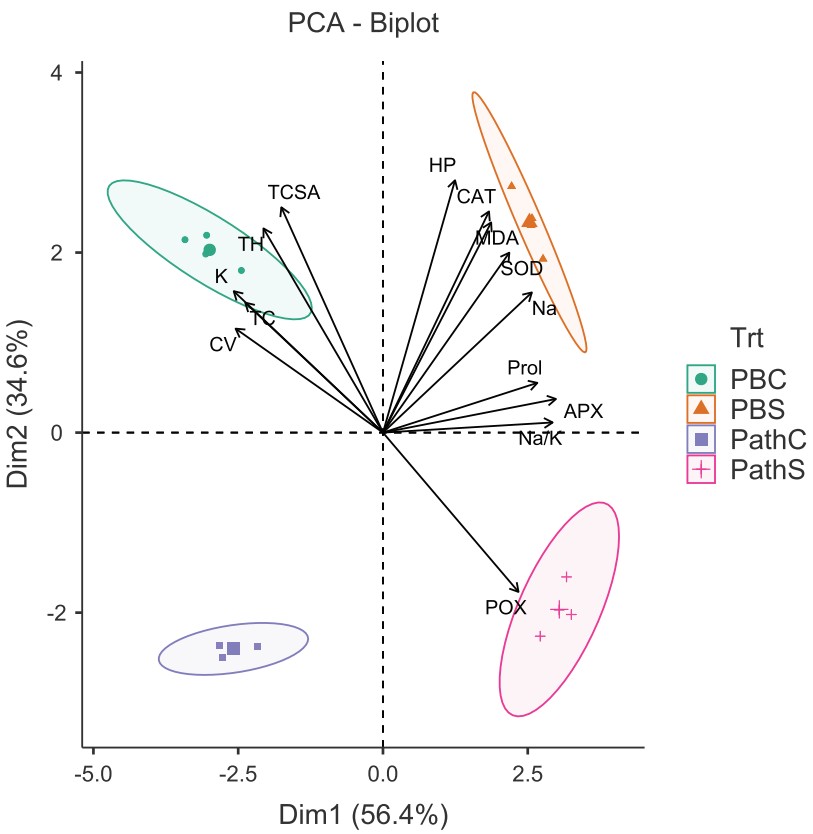

**Figure 1 Principal component analysis biplot showing variable loadings and cultivar-treatment groups on first two principal components in pear.** Lines radiating from the centre reflect relative contribution and directionality. PBC, Punjab Beauty control; PBS, Punjab Beauty sodic; PathC, Patharnakh control; PathS, Patharnakh sodic; Abbreviations: TH, tree height; TCSA, runk cross sectional area; CV, canopy volume; TC, total leaf chlorophyll; MDA, malondialdehyde; HP, hydrogen peroxide ($H_2O_2$); APX, ascorbate peroxidase; POX, peroxidase; CAT, catalase; SOD, superoxide dismutase; Na, leaf $Na^+$; K, leaf $K^+$; Na.K, leaf $Na^+/K^+$ ratio.

peach growth (*Thind & Mahal, 2020*) is anecdotal; key physiological traits underpinning sodicity tolerance remain elusive. In this backdrop, we aimed to understand the effects of sodicity stress on tree growth and physiological traits in commercially important pear and peach cultivars.

We found that peaches in general were more adversely affected than pears. Moreover, sodicity stress had a greater adverse effect on Patharnakh pear and Partap peach than other tested cultivars. The experimental soils used in this study still have pH >8.5, particularly below 30 cm depth. Additionally, a clayey sub-soil further limits the water flux and nutrient availability; hampering the plant growth (*Kumar et al., 2019*). This may partly explain the differential responses to sodicity stress of the crops and cultivar tested by us (*Samra, Singh & Sharma, 1988*). Excess salts suppress the vegetative growth and biomass production in pears (*Matsumoto et al., 2006*; *Okubo, Furukawa & Sakuratani, 2000*) and peaches (*Massai, Gucci & Tattini, 1997*; *Massai, Remorini & Tattini, 2004*); albeit in a genotype-specific manner such that some cultivars are more adversely affected than others (*Krishnamoorthy, 2009*). The depletion of leaf chlorophyll may be caused by the

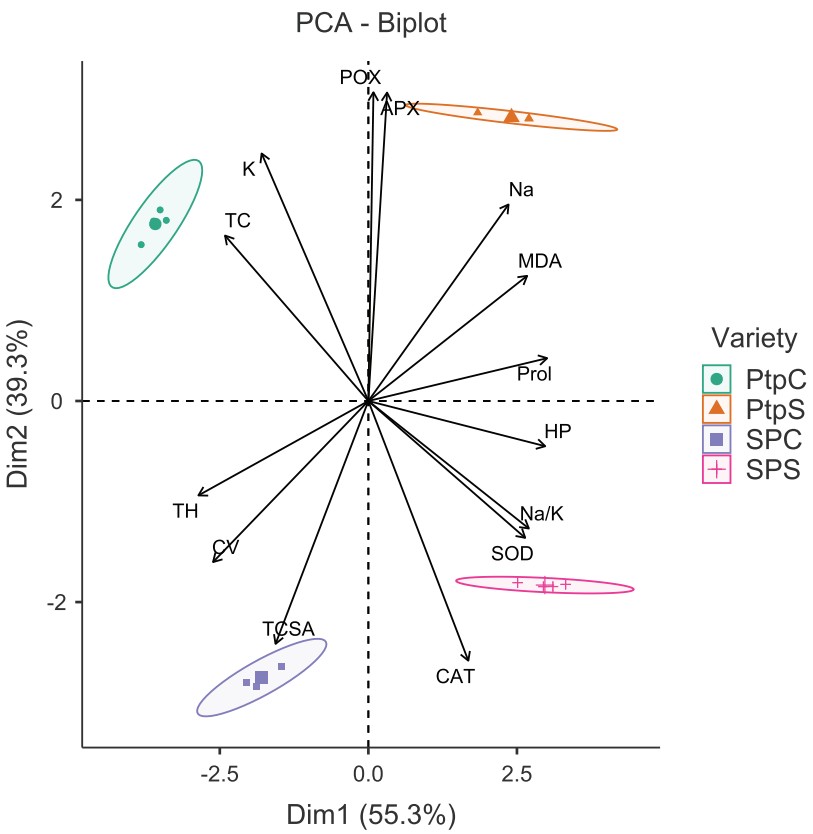

**Figure 2 Principal component analysis biplot showing variable loadings and cultivar-treatment groups on first two principal components in peach.** Lines radiating from the centre reflect relative contribution and directionality. PtpC, Partap control; PtpS, Partap sodic; SPC, Shan-e-Punjab control; SPS, Shan-e-Punjab sodic; Abbreviations: TH, tree height; TCSA, trunk cross sectional area; CV, canopy volume; TC, total leaf chlorophyll; MDA, malondialdehyde; HP, hydrogen peroxide ($H_2O_2$); APX, ascorbate peroxidase; POX, peroxidase; CAT, catalase; SOD, superoxide dismutase; Na, leaf $Na^+$; K, leaf $K^+$; Na.K, leaf $Na^+/K^+$ ratio.

lime-induced chlorosis when pears (*Ma et al., 2005*) and peaches (*Thomidis & Tsipouridis, 2005*) are grown in the high pH soils. The sodic soils of the study area are often deficient in iron (*Kaledhonkar, Meena & Sharma, 2019*), a key element in chlorophyll synthesis. Furthermore, high soil pH may inhibit the uptake of $Mg^{2+}$ and $Fe^{2+}$ ions needed for chlorophyll synthesis (*Jia et al., 2019*). The presence of bicarbonate ($NaHCO_3$, predominant salt in the study area soils; *Mandal, 2012*) suppresses iron availability to the peach (*Molassiotis et al., 2005*) and pear (*Valipour et al., 2020*) plants. The increase in apoplastic pH in the presence of bicarbonates limits iron transport to the root stele and restricts its uptake (*Molassiotis et al., 2005*).

Sodicity stress triggered a significant increase in leaf MDA and $H_2O_2$ accumulation, the major indicators of oxidative injury (*Sorkheh et al., 2012*; *Shen et al., 2021*); regardless of the crop and cultivar tested. Nonetheless, the increases in leaf MDA in sodicity-stressed plants relative to controls were much higher in peach (~37.0–42.0%) than in pear (9.61–19.91%) as were the increases in leaf $H_2O_2$ (~22.0–27.0% and ~12.0%, respectively), suggesting that peaches in general were more adversely affected by the oxidative stress
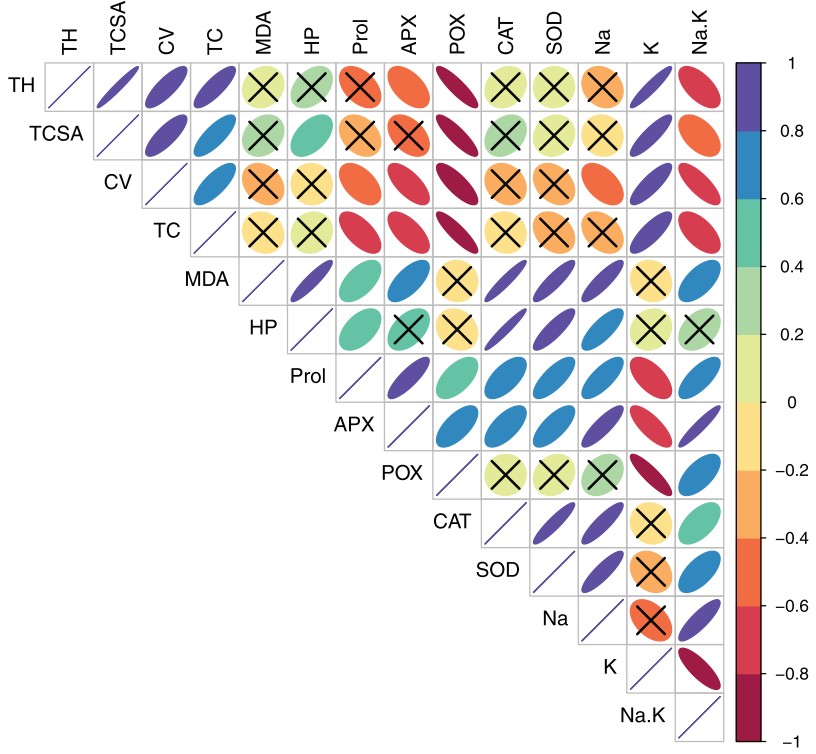

**Figure 3 Correlation plot showing Pearson's bivariate correlations between the measured traits in pear.** Ellipse size and color reflect the strength and direction (positive or negative) of the correlation. Individual cells marked with cross (X) denote non-significant correlations. Abbreviations: TH, tree height; TCSA, trunk cross sectional area; CV, canopy volume; TC, total leaf chlorophyll; MDA, malondialdehyde; HP, hydrogen peroxide ($H_2O_2$); APX, ascorbate peroxidase; POX, peroxidase; CAT, catalase; SOD, superoxide dismutase; Na, leaf $Na^+$; K, leaf $K^+$; Na.K, leaf $Na^+/K^+$ ratio.

(*Shahvali et al., 2020*). Although comparative studies are lacking, the *Prunus* spp. including peach are mostly highly sensitive to the salt-induced oxidative stress and lipid peroxidation (*Erturk et al., 2007*; *Shen et al., 2021*; *Toro, Pimentel & Salvatierra, 2021*). Salt-induced osmotic and oxidative stresses more adversely affect osmotic-sensitive than osmotic-tolerant genotypes; the latter show relatively efficient reactive oxygen species (ROS) scavenging when exposed to these stresses (*Rajabpoor et al., 2014*).

Although direct evidence is lacking, insights from a related species (*Pyrus betulaefolia* Bunge) suggest that salt-triggered increases in leaf MDA may not be deleterious enough to cause the lipid peroxidation in pears (*Wu & Zou, 2009*). The overexpression of genes '*Pp14-3-3*' (from *P. pyrifolia* (Burm F.) Nakai) and '*PbrNHX2*' (from *P. betulaefolia* Bunge) dramatically improved the salt tolerance in transgenic tobacco and *P. ussuriensis* Maxim by upregulating the activity of the antioxidant enzymes (*Li et al., 2014*; *Dong et al., 2019*). Specifically, increased APX activity may account for a better redox homeostasis in pear; '*14-3-3*' proteins interact with APX for scavenging the reactive oxygen species implicated in the oxidative damage (*Li et al., 2014*). Our results showed that sodicity-induced increases in APX activity were noticeably higher in pears (28.44–34.49%) than in peaches (13.49–29.41%). The cultivar differences for both MDA and $H_2O_2$ were highly significant
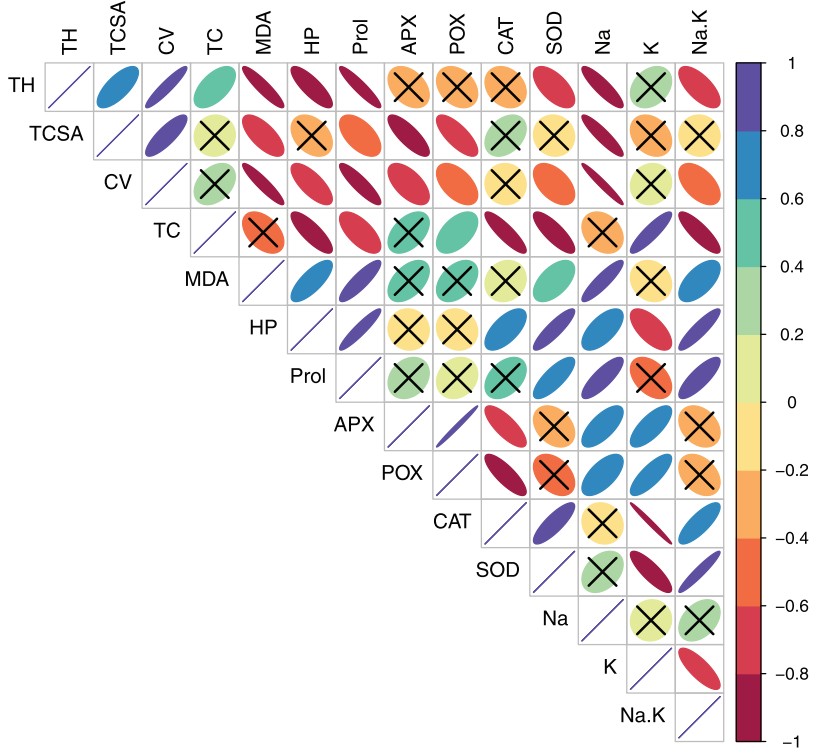

**Figure 4 Correlation plot showing Pearson's bivariate correlations between the measured traits in peach.** Ellipse size and color reflect the strength and direction (positive or negative) of the correlation. Individual cells marked with cross (X) denote non-significant correlations. Abbreviations: TH, tree height; TCSA, trunk cross sectional area; CV, canopy volume; TC, total leaf chlorophyll; MDA, malondialdehyde; HP, hydrogen peroxide ($H_2O_2$); APX, ascorbate peroxidase; POX, peroxidase; CAT, catalase; SOD, superoxide dismutase; Na, leaf $Na^+$; K, leaf $K^+$; Na.K, leaf $Na^+/K^+$ ratio.

($p < 0.001$) in pear (*Wu & Zou, 2009*), but only weekly significant (MDA: $p = 0.07$, $H_2O_2$: $p = 0.002$) in peach (*Toro, Pimentel & Salvatierra, 2021*); implying quite distinct and somewhat shared responses to sodicity stress of pear and peach cultivars, respectively, in terms of lipid peroxidation. Genotypic differences for these oxidative stress markers are either pronounced or subtle in different *Pyrus* and *Prunus* species (*Sorkheh et al., 2012*; *Rajabpoor et al., 2014*; *Zarafshar et al., 2014*; *Li et al., 2015*; *Toro, Pimentel & Salvatierra, 2021*).

Salt-stressed plants accumulate compatible osmolytes such as proline that, in addition to osmoregulation, also reduce the cellular damage by scavenging the ROS (*Dejampour et al., 2011*; *Rahneshan, Nasibi & Moghadam, 2018*; *Yousefi, Naseri & Zaare-Nahandi, 2019*). The tested cultivars did not differ significantly in leaf proline under both control and sodic treatments (*Wen et al., 2011*). Furthermore, the increase in leaf proline in response to MDA and $H_2O_2$ accumulation was moderate in pear ($r = 0.578$ and $0.519$, respectively; Fig. 3) but quite strong in peach ($r = 0.870$ and $0.930$, respectively; Fig. 4). Leaf proline levels may not always be sufficient enough to induce the osmotic adjustment (*Larher et al., 2009*; *Singh et al., 2022*; *Wen et al., 2011*). Despite being a typical osmolyte, proline may not essentially lessen the osmotic potential of pear leaves (*Larher et al., 2009*), and other
organic osmolytes such as glycine betaine may also be involved in the osmotic adjustment. The fact that MDA and $H_2O_2$ levels may not necessarily be toxic enough to upregulate the proline activity (*Wen et al., 2011*) also supports our findings as leaf MDA and $H_2O_2$ were two-three folds higher in peach than in pear (Table 2). Plausibly, a lower than expected increase in proline activity (*Regni et al., 2019*) might reflect higher sodicity tolerance in pear than in peach (*Ebert, 1999*; *Musacchi, Quartieri & Tagliavini, 2006*).

All the tested cultivars displayed increased activities of leaf antioxidant enzymes in response to sodicity stress. Enzymatic anti-oxidants efficiently protect salt-stressed plants from ROS induced oxidative stress, and are considered to be the reliable markers for discriminating the salt-tolerant and salt-sensitive genotypes (*Sorkheh et al., 2012*; *Yousefi, Naseri & Zaare-Nahandi, 2019*; *Aazami, Rasouli & Panahi, 2021*). The cultivar differences for anti-oxidant activities (*Regni et al., 2019*) can be explained by the complex nature of anti-oxidant expression in plant cells (*Racchi, 2013*) and the cell organelle-specific activities of anti-oxidant enzymes (*Niu & Liao, 2016*). The *P. pashia* rootstock was found to better protect the Flemish Beauty scions than clonal (Quince A and C) rootstocks against oxidative damage *via* enhanced CAT, POX and SOD activities (*Sharma & Sharma, 2008*). Similarly, peach seedling and clonal rootstocks differed considerably in leaf antioxidant levels in the presence of $NaHCO_3$ (*Molassiotis et al., 2005*). The SOD constitutes the first line of defense in alleviating the oxidative stress in plants; it dismutases the superoxide anion ($O_2^-$) to produce molecular oxygen ($O_2$) and $H_2O_2$. The CAT then decomposes $H_2O_2$ into $O_2$ and $H_2O$ (*Cavalcanti et al., 2004*). Obviously, a balance between SOD and CAT activities and not their relative levels *per se*, seem crucial to maintaining the $O_2^-$ and $H_2O_2$ levels in a steady-state (*Azarabadi et al., 2017*). The CAT and SOD activities were not only much higher (Table 3) but also had a clear synergistic effect (Figs. 1 and 2) in the sodicity-stressed Punjab Beauty pear and Shan-e-Punjab peach; enabling them to better adapt to the sodicity stress (*Sorkheh et al., 2012*). The decreased activity of CAT often comes at the expense of greater oxidative damage-characterized, for instance, by the increased accumulation of $H_2O_2$ (*Molassiotis et al., 2005*).

Sodicity-stressed pear and peach trees had significantly higher leaf $Na^+$ and lower $K^+$ than controls. In the sodic treatment, Punjab Beauty pear showed considerably lower increase in leaf $Na^+$ than Patharnakh; helping it maintain a higher leaf $K^+$. A more or less similar response was also seen in peach. Restricted translocation of $Na^+$ to aerial plant parts (*Matsumoto et al., 2006*), achieved for example by $Na^+$ exclusion in common pears (*Musacchi, Quartieri & Tagliavini, 2006*) prevents the xylem loading and translocation of $Na^+$ to the leaves. The differential accumulation of leaf $Na^+$ and $K^+$ in response to the salt stress has also been observed in both own-rooted and grafted peaches (*Massai, Gucci & Tattini, 1997*; *Massai, Remorini & Tattini, 2004*) and interspecific *Prunus* hybrids (*Dejampour et al., 2011*), with low $Na^+$ accumulators showing better salt tolerance (*Massai, Gucci & Tattini, 1997*). Reduced accumulation of leaf $Na^+$, achieved by root exclusion (*Musacchi, Quartieri & Tagliavini, 2006*) or partitioning into the basal leaves (*Massai, Remorini & Tattini, 2004*) together with the maintenance of adequate leaf $K^+$ (*Massai, Remorini & Tattini, 2004*) improves the salt tolerance.

In this study, the PCA was highly efficient in reducing the dimensionality, and in differentiating the cultivar- and sodicity-specific effects in data. Specifically, PCA delineated the putative traits linked to sodicity stress tolerance in the pear and peach cultivars. Previously, PCA has been used to unveiling the key responses to the salt stress in other fruit crops (*Sorkheh et al., 2012*; *Abid et al., 2020*). Multivariate techniques such as PCA are usually more suitable for detecting the key patterns in data having complex (multicollinear) variables (*Julkowska et al., 2019*). Additionally, graphical visualization of PCA loadings provides an easier and intuitive means to understanding the shared and contrasting physiological responses to the salt stress (*Sorkheh et al., 2012*; *Singh et al., 2022*). Based on the correlation analysis, MDA, $H_2O_2$ and leaf $Na^+$ were found to have a greater repressive effect on tree growth in peaches than in pears. Furthermore, a strong correlation between leaf $K^+$ and growth traits and leaf chlorophyll in pear, but not in peach, was indicative of leaf $K^+$ mediated osmotic adjustment in pears.

## CONCLUSIONS

Although sodicity stress suppressed tree growth regardless of the cultivar, strong genotypic differences were quite apparent: Punjab Beauty pear and Shan-e-Punjab peach exhibited better tolerance to sodicity stress. We found that sodicity-triggered increases in leaf malondialdehyde, hydrogen peroxide and $Na^+$ had a greater repressive effect on tree growth in peaches than in pears, and induced the proline-mediated osmotic adjustment in the former. The higher activities of catalase and superoxide dismutase enzymes coupled with restricted $Na^+$ uptake and the maintenance of adequate leaf $K^+$ are the plausible explanations for overall better sodicity tolerance in pear. Further studies should aim to understand the effects of sodicity stress on other biochemical and transcriptional changes, and how such changes influence the fruit yield and quality in the sodicity stressed pear and peach cultivars.

## ACKNOWLEDGEMENTS

We would like to thank the head of the Department of Fruit Science, Punjab Agricultural University, Ludhiana, India for providing the pear and peach plants. Mr. Dheeraj Kumar is appreciated for his technical help.

## ABBREVIATIONS

| | |
|---|---|
| **ANOVA** | Analysis of Variance |
| **APX** | Ascorbate peroxidase |
| **CAT** | Catalase |
| **CV** | Canopy volume |
| **DW** | Dry weight basis |
| **EC$_e$** | Soil saturation extract electrical conductivity |
| **H$_2$O$_2$** | Hydrogen peroxide |
| **MDA** | Malondialdehyde |
| **me L$^{-1}$** | Milli equivalent per liter |
| **PCA** | Principal Component Analysis |

| PCs | Principal Components |
|---|---|
| POX | Peroxidase |
| $pH_s$ | pH of saturated soil paste |
| ROS | Reactive oxygen species |
| SOD | Superoxide dismutase |
| TCSA | Trunk cross sectional area |
| TH | Tree height |

### Funding
This work was supported by Rashtriya Krishi Vikas Yojana, Govt. of Haryana, India. The funders had no role in study design, data collection and analysis, decision to publish, or preparation of the manuscript.

### Grant Disclosures
The following grant information was disclosed by the authors:
Rashtriya Krishi Vikas Yojana.

### Competing Interests
Anshuman Singh is an Academic Editor for PeerJ.

### Author Contributions
- Anshuman Singh conceived and designed the experiments, performed the experiments, analyzed the data, prepared figures and/or tables, authored or reviewed drafts of the article, and approved the final draft.
- Ashwani Kumar conceived and designed the experiments, performed the experiments, analyzed the data, prepared figures and/or tables, authored or reviewed drafts of the article, and approved the final draft.
- Parbodh Chander Sharma conceived and designed the experiments, authored or reviewed drafts of the article, and approved the final draft.
- Raj Kumar conceived and designed the experiments, prepared figures and/or tables, authored or reviewed drafts of the article, and approved the final draft.
- Rajender Kumar Yadav conceived and designed the experiments, authored or reviewed drafts of the article, and approved the final draft.

### Data Availability
The raw data is available in the Supplemental File.

### Supplemental Information
Supplemental information for this article can be found online at http://dx.doi.org/10.7717/peerj.14947#supplemental-information.

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
