# Peer review of "Sodicity stress differently influences physiological traits and anti-oxidant enzymes in pear and peach cultivars"

_PeerJ, doi:10.7717/peerj.14947_

## Round 0.1 · original submission · Major Revisions

Dear Authors
This MS is reviewed by the three reviewers and found that its need major revision.

Apart from reviewers comments the following comments may also consider during the revision

My additional suggested changes are:

1. Authors have described the effects of sodicity stress on fruit crops in two separate sentences (Lines 56-60). However, it is desirable to combine these two sentences for better coherence and meaning.

2. In the Introduction part, authors should briefly mention the rationale behind selecting the pear and peach cultivars, and should explicitly mention their adaptability and potential in the study region; citing relevant references.
3. Lines 64-66 read as “The physico-chemical properties of sodic soils improved following amendment application and salt leaching; however, such improvements are mostly transient and limited to the top soil”. It would be better if it is revised as: Although sodic soils display noticeable improvements in the physico-chemical properties following amendment application and salt leaching with fresh water, such improvements are mostly transient and confined to the top soil.

4. In the ‘experimental material’ section (lines 109-111), the scientific names should be properly written. Pyrus communis should be written as Pyrus communis L., Pyrus pyrifolia  as Pyrus pyrifolia (Burm F.) Nakai, Pyrus pashia as Pyrus pashia Buch.-Ham. Ex D. Don, and Prunus persica as Prunus persica (L.) Batch.

5. Lines 117-118 mention irrigation. The interval of irrigation (for example, days or weeks) should be given.

6. Although authors have used the effect size measure (lines 172-173), suitable logic has not been provided for using the same. Thus, authors should support the use of effect-size measure with suitable reference.

7. In the Results section, the ANOVA results need to be re-written. Specifically, lines 186-190 need to be revised for more clarity and a better meaning.

8. Wherever ‘sodic’ and ‘control’ treatments have been mentioned in the Results section, they should be followed by their respective soil pHs values: e.g., in sodic (soil pH s ~8.8) than in control treatment (pH s ~8.2).

9. In lines 316 and 318, Pyrus betulifolia should be written as Pyrus betulifolia Bunge. 10. In line 319, Pyrus ussuriensis should be written as Pyrus ussuriensis Maxim.

11. In the main manuscript, hydrogen peroxide has been written as H 2 O 2, but in Supplementary Tables as HP; which needs to be corrected in all the Supplementary Tables.

Reviewer 1 ·

Basic reporting

The manuscript has been written in a distinct and apparent way, and the findings are discussed befitting manner. The English language seems up to the mark. The literature reviewed by the authors is adequate and there is a meticulous discussion of the results.

Experimental design

The authors have used appropriate design and statistical tests.

Validity of the findings

The findings reported in this investigation are quite noteworthy and meaningful to the future researches in the similar areas. The findings are prop-up by enough supporting data.

Additional comments

1. In the Methods section of Abstract, write ‘physiological traits of’ and not ‘physiological traits on’
2. In lines 39-40, authors have mentioned ‘saturated soil paste pH’. Instead, it should be mentioned as the ‘pH of saturated soil paste’ here and in other places (for example line 122).
3. The sentence in lines 70-71, needs to be revised as: development of salt tolerant scion and root-stock cultivars is absolutely essential to sustain fruit production in the salt-affected soils.
4. In the lines 75-76, revise as: tree growth and physiology of pears and peaches.
5. In the line 78, write as: soil pH and related sub-soil constraints may suppress plant growth.
6. The scientific names used throughout should be followed by the authority name: for example as Prunus persica (L.) Batch.
7. Lines 114 to 116 mention about the planting distance adopted. However, no reference has been provided here. Thus, suitable reference needs to be provided for both pears and peaches.
8. Rewrite the line 167-169 as: The main and interaction effects of independent factors (treatment and cultivar) on each dependent variable were examined by a two-way Analysis of Variance.
9. In the line 174, it seems that authors have erroneously mentioned as standard error, but in Tables it is invariably given as SD (standard deviation). So, this needs to be corrected.
10. In the line 194 of the Results, ‘of peaches compared to pears’ needs to be revised to ‘of peaches than of pears’.
11. In the results section (e.g., subheading Tree Growth) and in rest of the manuscript wherever applicable, write ‘cultivar’ and not ‘cv.’.
12. Lines 268-269 in the Correlation analysis should be revised for a better meaning.
13. The lines 279-281 “ We found…………their counterparts’ needs to be revised for a better sense.
14. In line 352, write as ‘and are considered to be’.

·

Basic reporting

1. Current reference available? 2022 indtead of 2018
Horticultural production will have to increasingly rely on salt-impaired lands in the coming 37 decades (Singh et al., 2018a)
2. If it is updated add latest one instead of 2011 data
Saline, sodic (alkali) and saline-sodic soils occupy ~60%, 26% and 38 14% of the global salt-affected area, respectively (Wicke et al., 2011).
3. Support introduction and discussion with latest reference and delete very old one. For eg:
The Peach (Prunus persica) CBL and CIPK Family Genes ( K Qiu • 2022);


(PDF) The Behavior of Peach and Pear Trees under Extreme ...
https://www.researchgate.net › publication › 288012884_...

21-Sept-2022

4. For reference check for uniformity
5. Language must be improved.
6. Table-3 check unit for superscript and subscript
7. Figures legends must be explanatory (significant findings).

Experimental design

Appropriate

Validity of the findings

appropriate

Additional comments

already submitted in first section

·

Basic reporting

1. The full forms or the abbreviations used in the MS needs to be mentioned on their first mention. Rectify throughout the MS.
2. The introductory sentence in the introduction section is not appropriate. Redraft the introduction accordingly.
3. Highlight the novelty of the study.
4. The methodology for Leaf Na+ and K+ needs to be elaborated for replication of experiment.
5. The discussion has been excessively elaborated. Redraft the same to shorten it.
6. Concluding remark for each of the activity for results may be added for clarity of the results.
7. The future prospects of the study may be added.

Experimental design

The experimental section may be elaborated for replication of experiment.

Validity of the findings

The findings have been fairly validated.

---

## Round 0.2 · Minor Revisions

Please revise the comments of the reviewers and resubmit for consideration.

Reviewer 1 ·

Basic reporting

No comment

Experimental design

No comment

Validity of the findings

No comment

Additional comments

Author have rectified the indicated comments on manuscript. I am satisfy with given justification

·

Basic reporting

The authors have adequately addressed all the comments. However, the full forms or the abbreviations used in the MS needs to be mentioned on their first mention. This needs to be rectified throughout the MS.
Also the English of the MS may be improved.

Experimental design

Adequately addressed

Validity of the findings

Adequate

---

## Round 0.3 · accepted · Accept

All the comments have been resolved properly now the manuscript is ready for publication